# Cost-effectiveness of monitoring and liver cancer surveillance among patients with inactive chronic hepatitis B

Mehlika Toy[1]*, David Hutton[2], Erin E. Conners[3], Hang Pham[4], Joshua A. Salomon[5,6], Samuel So[4]

1 Erasmus School of Health Policy & Management, Erasmus University, Rotterdam, The Netherlands, 2 Department of Health Management and Policy, University of Michigan, Ann Arbor, Michigan, United States of America, 3 Division of Viral Hepatitis, Centers for Disease Control and Prevention, Atlanta, Georgia, United States of America, 4 Department of Surgery, Asian Liver Center, Stanford University School of Medicine, Stanford, California, United States of America, 5 Department of Health Policy, Stanford University School of Medicine, Stanford, California, United States of America, 6 Center for Health Policy, Freeman Spogli Institute for International Studies, Stanford University, Stanford, California, United States of America

* toy@eshpm.eur.nl

**Data Availability Statement:** All relevant data are within the manuscript and its Supporting information files.

## Abstract

Patients with chronic hepatitis B infection (CHB) have an increased risk for death from liver cirrhosis and hepatocellular carcinoma (HCC). In the United States, only an estimated 37% of adults with chronic hepatitis B diagnosis without cirrhosis receive monitoring with at least an annual alanine transaminase (ALT) and hepatitis B deoxyribonucleic acid (DNA), and an estimated 59% receive antiviral treatment when they develop active hepatitis or cirrhosis. A Markov model was used to calculate the costs, health impact and cost-effectiveness of increased monitoring of adults with HBeAg negative inactive or HBeAg positive immune tolerant CHB who have no cirrhosis or significant fibrosis and are not recommended by the current American Association for the Study of Liver Diseases (AASLD) clinical practice guidelines to receive antiviral treatment, and to assess whether the addition of HCC surveillance would be cost-effective. For every 100,000 adults with CHB who were initially not recommended for treatment, if the monitoring rate increased from the current 37% to 90% and treatment rate increased from 59% to 80%, 4,600 cases of cirrhosis, 2,450 cases of HCC and 4,700 HBV-related deaths would be averted with a gain of 45,000 QALYs and a savings of $180 million in lifetime health care costs. At a willingness to pay threshold of $100,000/ QALY, the addition of HCC surveillance with the standard recommended biannual liver ultrasound and alfa fetoprotein levels is likely cost-effective if the HCC risk $\geq$ 0.55%/year. Regular monitoring of persons with inactive or immune tolerant CHB who are initially not recommended to receive antiviral treatment in the United States is cost-saving. The addition of HCC surveillance with biannual US and AFP would be cost-effective for individuals with HCC incidence $\geq$ 0.55%/year.

**Funding:** This work was supported by the Centers for Disease Control and Prevention, National Center for HIV, Viral Hepatitis, STD, and TB Prevention Epidemiologic and Economic Modeling Agreement [NEEMA, NU38PS004651].

**Competing interests:** NO authors have competing interests.

## Introduction

In the United States, there are an estimated 1.25 to 2.4 million HBsAg-positive persons living with chronic hepatitis B (CHB) infection [1, 2]. Most of them have no or few symptoms and less than 50% are aware of their infection [3]. A significant proportion of adults tested positive for HBsAg are initially not recommended for antiviral treatment according to the AASLD clinical practice guidelines because they have HBeAg negative inactive CHB (anti-HBe positive, persistently normal ALT, HBV DNA < 2,000 IU/mL, and with no or minimal liver necroinflammation or fibrosis) or HBeAg positive immune tolerant HBV infection (normal or minimally elevated ALT, very high HBV DNA, and minimal liver inflammation and no fibrosis) [4]. Although antiviral treatment is not currently recommended for inactive or immune tolerant CHB, AASLD recommends regular monitoring of ALT and HBV DNA levels to determine whether an indication for treatment has developed, and HBeAg testing is also recommended in HBeAg positive patients to assess when they would become e antigen negative [4]. Despite these recommendations, reported monitoring of ALT and HBV DNA among adults with CHB remains low [5]. The estimated monitoring rate of adults in the U.S. with CHB with at least an annual ALT and HBV DNA in individuals without cirrhosis is only 37%, with an estimated 59% receiving antiviral treatment when they become eligible for treatment [3].

The AASLD also recommends HBsAg positive individuals with an annual HCC risk greater than 0.2% per year to receive hepatocellular carcinoma (HCC) surveillance with biannual liver ultrasound with or without blood levels of alfa-fetoprotein (AFP). Despite this recommendation, only 36% of CHB without cirrhosis undergo HCC surveillance [5]. The recommended threshold for HCC surveillance was largely based on an unpublished cost-effectiveness study from 1999 that reported HCC surveillance with biannual ultrasound and AFP in a cohort of hepatitis B carriers ≥ age 30 years with an annual HCC incidence ≥ 0.2% would prolong life by 3 months and would be cost-effective at a willingness to pay at $50,000/life year gained [6, 7]. Although recent studies reported HCC surveillance in patients with cirrhosis is likely cost-effective, the cost-effectiveness of HCC surveillance in non-cirrhotic chronic hepatitis B patients in the U.S. have not been re-examined.

The aim of this study is to assess 1) the current population level costs, cost-effectiveness and health impact of monitoring of HBsAg-positive persons who are currently not on antiviral treatment, and 2) the threshold in HCC incidence for surveillance to be cost-effective in the United States.

## Methods

A Markov model was developed using TreeAge Pro 2023 to simulate long-term outcomes including cirrhosis, HCC, and CHB-related death as patients with HBeAg negative inactive CHB and HBeAg positive immune tolerant CHB move through various health states (Fig 1). Once patients enter the HBeAg positive immune active or HBeAg negative active CHB or cirrhosis health state they become candidates for treatment following the latest 2018 AASLD clinical practice guidelines [4]. Individuals who receive treatment for active CHB and cirrhosis have a lower risk of developing liver-related complications such as HCC and cirrhosis following disease progression rates derived from cohort studies and meta-analyses of HBV mono-infected patients (S1 and S2 Tables in S1 File). Without monitoring and antiviral treatment, patients follow the disease natural history. The natural history of CHB and disease progression rates were derived from recent cohort studies and meta-analyses mainly from North America for HBV mono-infected patients (S1 Table in S1 File). Disease progression rates were assumed to be 50% lower among women compared to men, based on recent sex-specific studies [8–10]. Treatment effectiveness estimates were expressed as reductions in disease progression risks for

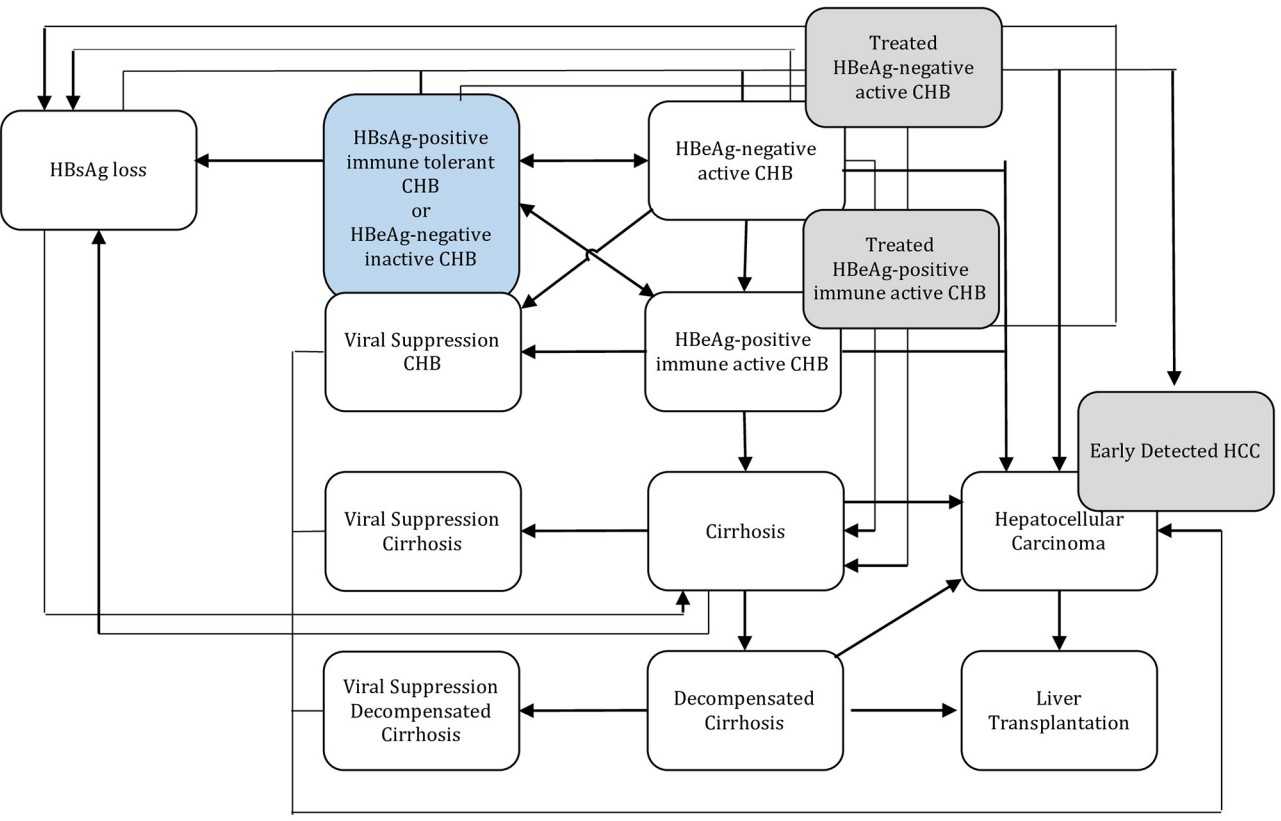

**Fig 1. Markov schematic.**

treatment naïve patients (S2 Table in S1 File). We assumed effective antiviral suppression would reduce liver cancer risks in cirrhotic and non-cirrhotic patients by 50% and 70% respectively, compared with natural history [11, 12]. Disease progression between health states, conditional on treatment, age (where available) and sex was simulated in one-year cycles. Causes of death that were not related to CHB were included in the model, based on age-specific mortality rates from life tables in the National Statistics Report [13]. Annual probabilities of receiving a liver transplant for CHB-related decompensated cirrhosis and HCC (1.2% and 7%, respectively) were calculated based on data from Organ Procurement and Transplantation Network [14]. If progression rates were reported, these were transformed into annual probabilities using a standard formula ($P = 1-e^{-rt}$), where P is the probability and r is the annualized progression rate.

We ran the model for representative ages of 25, 35, 45, 55, 65, and 75 to represent the age groups of 18–29, 30–39, 40–49, 50–59, 60–69, 70+. We then weighted these results by U.S. Census data by age and aggregated across age cohorts to create a hypothetical cohort of 100,000 adults. Key input variables are shown in Table 1.

## Scenarios

Three scenarios were analyzed in the model. 1). *Current practice (CP)* with 37% of CHB patients who are initially not eligible for treatment receiving monitoring with biannual ALT and annual HBV DNA [3] and 59% who become treatment eligible receiving antiviral treatment. 2). *Increased monitoring and treatment (increased M&T)* where the percentage of

**Table 1. Key input variables.**

| Variable | Base Case | Range | References |
|---|---|---|---|
| Age/birth cohort | ≥18 yrs | 18–70 yrs | |
| Male to female ratio of positive HBsAg population | 58:42 | | Patel et al. 2019 [15] |
| Percent receiving monitoring (current practice) | 37% | 20%-60% | Spradling et al. 2016 [16] |
| Percent receiving treatment if eligible (current practice) | 59% | 10%-90% | Spradling et al. 2016 [16] |
| Reduction in mortality risk with HCC surveillance | 38% | | Mittal et al. 2016 [17] |
| **Linkage to and Treatment Costs** | | | |
| Antiviral drug costs per year[a] | $387 | $325-$16,464[b] | Redbook (May, 2024) [18] |
| Total annual monitoring costs (without HCC surveillance cost)‡ | $221 | $111–332 | Medicare reimbursement |
| Clinic visit x 2 | $74 | $37-$111 | Medicare reimbursement |
| ALT x 2 | $7 | $4-$11 | Medicare reimbursement |
| HBV DNA x 1 | $59 | $29-$88 | Medicare reimbursement |
| HBeAg | $11.53 | $10–13 | Medicare reimbursement |
| Total annual HCC surveillance costs[c] | $296 | $100-$554 | Medicare reimbursement |
| Ultrasound x 2 | $125 x2 | ($62-$187)x2 | Medicare reimbursement |
| AFP x 2 | $23 x 2 | ($12-$35)x2 | Medicare reimbursement |
| Annual Disease Management Costs[d] | | | |
| Chronic Hepatitis B | $2,002 | $202-$7,816 | Liu et al. 2012 [19] |
| Cirrhosis | $5,964 | $202-$7,096 | Liu et al. 2012 [19] |
| Decompensated cirrhosis | $15,795 | $4,901-$37,081 | Liu et al. 2012 [19] |
| Symptom detected HCC | $99,207 | $77,076-$115,614 | Parikh et al. 2020 [20] |
| Screen detected HCC non-cirrhosis | $41,360 | $32,134-$48,200 | Parikh et al. 2020 [20] |
| Screen detected HCC cirrhosis | $87,789 | $68,205-$102,307 | Parikh et al. 2020 [20] |
| Liver Transplantation 1st year | $215,162 | $167,163-$250,746 | Liu et al. 2012 [19] |
| Liver Transplantation 2nd year | $26,860 | $23,958-$35,937 | Liu et al. 2012 [19] |
| **Health State Utilities** | | | |
| Active CHB | 0.91 | (0.80–0.92) | Woo et al. (EQ-5D) [21] |
| Cirrhosis | 0.88 | (0.78–0.88) | Woo et al. (EQ-5D) [21] |
| Inactive CHB | 1.00 | (0.90–1.00) | Assumption |
| Decompensated cirrhosis | 0.73 | (0.49–0.82) | Woo et al. (EQ-5D) [21] |
| Symptom detected HCC | 0.67 | (0.54–0.80) | Parikh et al. 2020 [20] |
| Screen detected HCC non-cirrhosis | 0.81 | (0.65–0.85) | Woo et al. (EQ-5D) [21] |
| Screen detected HCC cirrhosis | 0.70 | (0.66–0.84) | Parikh et al. 2020 [20] |
| Liver Transplantation | 0.84 | (0.72–0.84) | Woo et al. (EQ-5D) [21] |
| HBsAg seroclearance | 1.00 | (0.95–1.00) | Assumption |
| Viral suppression | 1.00 | (0.95–1.00) | Assumption |

[a]Assuming 60% on generic TDF and 40% on generic ETV [22].

[b]This is the range for one-way sensitivity analysis, but for the probabilistic sensitivity analysis, it varies from $325 to $16,460.

[c]Annual monitoring is the total cost including 2x clinic visit, 2x ALT and 1x HBV DNA level as recommended by AASLD [23]. Annual HCC surveillance cost includes a scenario with a clinical visit and 2x US and 2x AFP, a scenario with 1x US and 2x AFP, and a scenario with 2x US and no AFP.

[d]Adjusted to 2024 USD using the Medical Consumer Price Index (CPI) calculator [24].

Abbreviations: AFP, alpha fetoprotein; ALT, alanine aminotransferase; CBC, complete blood count; CHB, chronic hepatitis B; HBeAg, hepatitis B e antigen; LFT, liver function tests; OPTN, Organ Procurement and Transplantation Network.

patients receiving monitoring and treatment increased to 90% and 80%, respectively. 3). *M&T plus HCC surveillance*, with increased monitoring and treatment of 90% and 80%, respectively, and 3 different HCC surveillance strategies to assess at what HCC incidence would the addition of HCC surveillance would be cost-effective: biannual liver ultrasound+AFP

recommended by AASLD, biannual ultrasound recommended by the European Association for the Study of the Liver (EASL) and an annual ultrasound+biannual AFP which likely reflects the reality in most clinical practice. When we evaluate these different surveillance strategies, we assume they have equivalent reductions in liver cancer-related mortality but have different costs (Table 1).

In each scenario, we assumed among patients who received monitoring and treatment, adherence was 90%, and patients who did not receive monitoring followed the natural history. In the HCC surveillance scenarios, we assumed 70% of the patients received HCC surveillance.

## Costs and utilities

The costs of annual monitoring for HBeAg negative inactive CHB ($221) was based on the Medicare reimbursement costs for bi-annual clinic visits, bi-annual blood tests for ALT and an annual HBV DNA (Table 1). The costs of annual monitoring for the estimated 26% [25] of CHB patients who are HBeAg positive would include an additional HBeAg test ($11.53). The cost of HCC surveillance was based on Medicare reimbursement of $125 for a single liver ultrasound and $23 for a single AFP test. The estimated annual medical management costs of surveillance-detected HCC (assuming 60% were early stage and 40% were intermediate or late stage HCC) among non-cirrhotic patients was $40,167 and $85,256 among cirrhotic patients, as derived from Parikh et al. [20]. The estimated annual medical management costs of symptom detected HCC was $96,345. We assume patients who meet treatment eligibility will be prescribed generic tenofovir disoproxil fumarate (TDF) or generic entecavir in the United States. Although the lowest price for generic TDF at $325 per year [26], we used an annual antiviral drug cost of $387 assuming 60% of the patients will be dispensed generic TDF and 40% generic ETV ($480 per year) [22]. Other medical management costs for CHB, cirrhosis, decompensated cirrhosis, and liver cancer were obtained from Liu et al. [19]. Medical management costs were adjusted for inflation using the US consumer price index to reflect 2024 US dollars [24]. We assumed patients who achieved HBsAg loss would continue to incur annual costs for long-term CHB monitoring. We added age-specific per person background medical costs to our analysis using a recently published US catalogue [27]. All costs and QALYs were discounted at a rate of 3% per year. The analysis was performed from the health care system perspective. We used EQ5D utility assessments calculated by Woo et al. [21] based on a Canadian CHB patient sample and included age adjustments from Parikh et al. [20]. All key input variables related to costs and utilities are shown in Table 1.

## Sensitivity analysis

One-way sensitivity analysis was used to determine the parameters that had the greatest impact on the results. We also reported the findings when only monitoring was increased without an increase in treatment, and an increase in treatment rate without an increase in monitoring, and adherence to monitoring and treatment from 60% to 90%. In the base case, when we define a percentage of monitoring, we mean that a certain percentage of the population is monitored regularly for the rest of their lives and the rest of the population receives no monitoring for the rest of their lives. However, in an alternative sensitivity analysis scenario, we alter this definition so that we assume that all individuals are receiving monitoring, but at irregular rates. Under this alternative scenario, the definition of the percentage of monitoring is the fraction of the population receiving monitoring in any particular year, and this may not be the same individuals necessarily monitored each year. To assess the cost-effectiveness of HCC surveillance among patients with inactive and immune tolerant CHB, annual HCC incidence from 0.1% to 1.0% was examined. Finally, we conducted a probabilistic sensitivity

**Table 2. Cost-effectiveness and health outcomes of increased monitoring of adults with HBeAg-negative inactive CHB and HBeAg-positive immune tolerant CHB to 90% and increased antiviral treatment to 80% when they become treatment eligible compared to current practice for a population of 100,000 adults with inactive or immune tolerant CHB.**

| Scenario | Cost | QALYs | ICER | Cirrhosis | Decompensated Cirrhosis | HCC | HBV Deaths |
|---|---|---|---|---|---|---|---|
| Current Practice (CP) | $18,275,900,000 | 1,891,800 | - | 17,210 | 2,090 | 8,470 | 12,100 |
| Increased M&T | $18,257,900,000 | 1,936,900 | Cost-saving | 12,620 | 590 | 6,020 | 7,400 |
| Difference | -$18,000,000 | +45,100 | -$399 | -4,590 | -1,500 | -2,450 | -4,700 |

analysis varying all parameter values across specified distributions to evaluate the impact of overall parameter uncertainty on outcomes.

## Results

### Increasing monitoring and treatment

Compared to current practice, where an estimated 37% of persons with inactive or immune tolerant CHB are monitored and 59% receiving antiviral therapy when they become treatment eligible, increased monitoring to 90% and treatment to 80% (*increased M&T*) would be cost-saving (Table 2). In the current practice scenario, among the cohort of adults with inactive or immune active CHB, an estimated 17.21% would develop cirrhosis, 2.09% would develop decompensated cirrhosis, 8.47% would develop HCC, and 12.10% would die from CHB-related causes. For every 100,000 persons with inactive or immune active CHB, increasing monitoring to 90% and antiviral treatment for eligible individuals to 80% would save $180 million in lifetime health care costs, add 45,100 QALYs and would avert 4,590 cases of cirrhosis, 1,500 cases of decompensated cirrhosis, 2,450 cases of HCC, and 4,700 HBV related deaths. The decomposition of the total costs is presented in S1 Fig.

### HCC surveillance

Adding HCC surveillance to increased monitoring and treatment in the cohort of inactive or immune tolerant CHB patients regardless of annual HCC risk would not be cost-effective ($266,728/QALY compared with current practice). At a willingness to pay at $100,000 per QALY, the incremental cost-effectiveness ratio (ICER) for increased M&T plus the AASLD recommended HCC surveillance strategy with biannual US and AFP ($296/year) would become cost-effective at HCC incidence $\geq$ 0.55%/year (Fig 2A). The ICER for HCC surveillance strategy with an annual ultrasound and biannual AFP ($171/year), would become cost-effective at HCC incidence $\geq$ 0.30%/year (Fig 2B). The ICER for the European Association for the Study of the Liver (EASL) recommended HCC surveillance strategy (biannual US without AFP ($250/year) would become cost-effective at HCC incidence $\geq$ 0.39%/year (Fig 2C).

### Sensitivity analysis

In the scenario where the definition of the percent monitoring assumes all individuals with inactive and immune tolerant CHB potentially could be monitored, but not the same individuals regularly receiving monitoring each year, increasing the monitoring to 90% is considered cost-effective at $24,951 per QALY gained, but no longer cost-saving (S3 Table in S1 File). Under this assumption, outcomes for the status quo intervention are better because all people eventually get exposed to monitoring and enter treatment when an indication for treatment has developed, albeit at a slower rate. Increasing the monitoring rate increased the likelihood that patients who developed active hepatitis or cirrhosis would receive treatment. Although

A
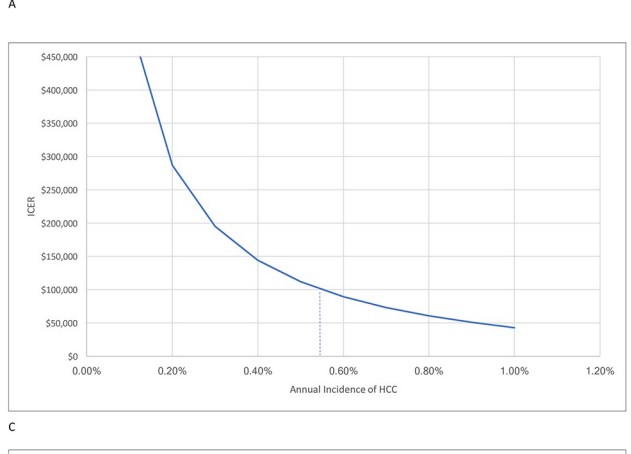

B
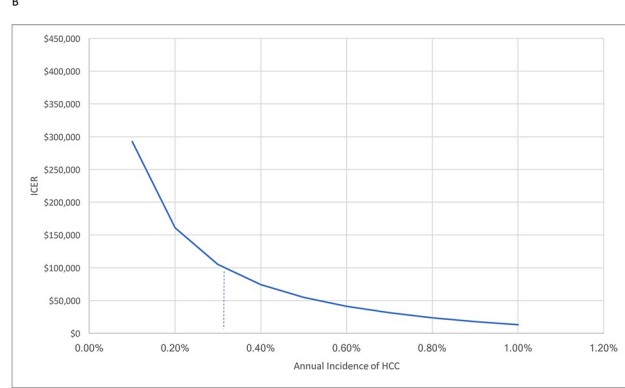

C
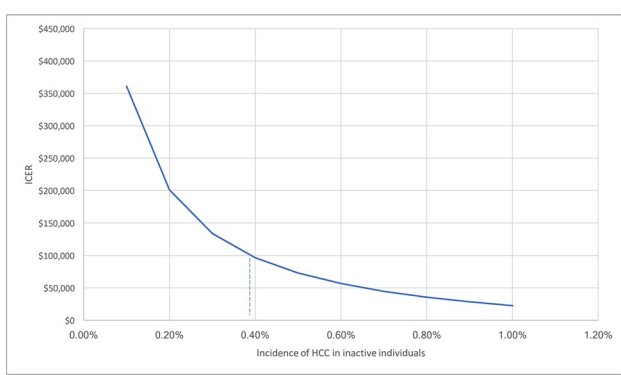

**Fig 2. A**. Cost-effectiveness based on incidence of HCC in individuals not currently on antiviral treatment with biannual liver ultrasound and AFP (AASLD guidelines). * At a willingness to pay at $100,000 per QALY, HCC surveillance with biannual liver ultrasound and AFP would be cost-effective if HCC risk is ≥ 0.55%/year in HBsAg-postive individuals who are not currently on antiviral treatment. **B**. Cost-Effectiveness based on incidence of HCC in individuals not currently on antiviral treatment with annual liver ultrasound and biannual AFP. * At a willingness to pay at $100,000 per QALY, HCC surveillance with annual liver ultrasound and biannual AFP would be cost-effective if HCC risk is ≥ 0.3%/year in HBsAg-postive individuals who are not currently on antiviral treatment. **C**. Cost-Effectiveness based on incidence of HCC in individuals not currently on antiviral treatment with biannual liver ultrasound (EASL guidelines). * At a willingness to pay at $100,000 per QALY, HCC surveillance with biannual liver ultrasound redwould be cost-effective if HCC risk is ≥ 0.39%/year in HBsAg-postive individuals who are not currently on antiviral treatment.

the incremental benefits under this assumption are lower, the increased M&T strategy would still be highly cost-effective.

The probabilistic sensitivity analysis which varies all parameters simultaneously shows a > 99% likelihood that the increased M&T strategy would be cost-effective at a willingness to pay threshold of $100,000/QALY (S2 Fig). S3 Fig shows the results of one-way sensitivity analysis summarized using a tornado plot. The model was sensitive to a few parameters like discount rate on utility, HCC incidence, cost of antiviral therapy and utility of inactive CHB. Increasing monitoring has more health gains compared to increasing treatment alone (S4A and S4B Table in S1 File). For adults with HCC incidence ≥ 0.55%/year, increased M&T plus AASLD recommended HCC surveillance with US+AFP every 6 months yields 20,602,000 life years, which compared to the M&T scenario alone, HCC surveillance has an additional 400 life years gain for a 100,000 population (S5 Table in S1 File).

## Discussion

Although adults with immune tolerant CHB or inactive CHB are not recommended by the current AASLD clinical practice guidelines to receive antiviral therapy, this study found

regular monitoring with biannual blood tests for disease activity and antiviral treatment when treatment becomes indicated would be highly cost-effective.

The cost-effectiveness of HCC surveillance among non-cirrhotic CHB patients have not been re-examined since 1999 when Collier, Krahn and Sherman, in an unpublished study, reported HCC surveillance with biannual liver ultrasound and AFP would be cost-effective (at a willingness to pay at $50,000 per life year gained) in HBV carriers over 30 years of age with an annual HCC risk of 0.2% or greater and would prolong life by 90 days. AASLD has adopted this threshold and recommends HCC surveillance with biannual ultrasound and AFP for patients with CHB with an annual HCC risk of 0.2% or greater. Accordingly, AASLD recommends HCC surveillance for CHB patients with cirrhosis, Asian male over 40 years, Asian female over 50 years (with estimated HCC incidence 3–8%/yr, 0.4–0.6%/yr, and 0.3–0.6%, respectively), as well as CHB patients with family history of HCC and African and North American Black.

The current study found at a willingness to pay at $100,000/QALY, HCC surveillance with biannual liver ultrasound and AFP in adults with CHB would not be cost-effective at a willingness to pay at $100,000/QALY unless the annual HCC risk is ≥ 0.55% which is 2.75 times higher than the threshold which was reported by Collier and adopted by AASLD. HCC surveillance with biannual ultrasound alone that is recommended by EASL would become cost-effective for patients with annual HCC incidence ≥ 0.40%.

Although EASL recommend only biannual liver ultrasound without AFP for HCC surveillance, a meta-analysis showed the addition of AFP measurements can increase the sensitivity of detecting early stage HCC to 63% compared with 45% with ultrasound alone [28]. Recent studies also found a rising AFP level on serial measurements improved the sensitivity of AFP to detect HCC to 77.1% - 87.5% in patients with cirrhosis or advanced fibrosis [29]. In clinical practice, many providers found compliance among adults with CHB to obtain more than one ultrasound a year is poor. There is no evidence that there is a survival difference between HCC diagnosed with annual or biannual liver ultrasound [30, 31]. In this study, we found an alternate HCC surveillance strategy with an annual liver ultrasound and biannual AFP would be cost-effective for CHB patients with an annual HCC risk as low as 0.3%.

In a meta-analysis of 59 studies reported between 2014 and 2019, biannual HCC surveillance in patients with liver cirrhosis is generally associated with improved detection at an earlier stage, increased chance for potentially curative treatment and prolonged survival [32]. Whether HCC surveillance leads to an overall decrease in patient mortality remains an ongoing debate because of the lack of clinical trial studies. Although the risk of HCC is greatest in patients with cirrhosis, patients with CHB without cirrhosis can also develop HCC especially if there is a family history of a first degree relative with HCC.

Our study could be used as additional evidence to inform updated guidelines. With that said, our estimates of the precise threshold for initiating HCC surveillance among adults with inactive and immune tolerant CHB are uncertain and dependent upon assumptions about the benefits of early HCC treatment. To the extent that surveillance may lead to early, inexpensive treatment and reduce the need for increasing use of expensive immunotherapies to treat late stage HCC, surveillance may become more cost-effective.

Our study has a few limitations. We have obtained the 37% of CHB patients receiving at least an annual ALT and HBV DNA and 59% percent receiving treatment if eligible from the The Chronic Hepatitis Cohort Study (CHeCS) study by Spradling et al. [16]. This study might not be generalizable for the CHB population in the United States, since the CHeCS study attempts to link to care in their system all those found to have CHB. Also, each of the large health research centers and hospitals that participated in the CHeCS study have liver specialists that are experts at managing CHB. Other studies based on insurance claims data reported 36%

-50% of CHB patients without cirrhosis received at least an annual ALT and HBV DNA or HBeAg [22, 33]. Wong et al found only 37.3% of treatment eligible CHB patients in four large safety-net health systems received antiviral treatment [34]. We also made assumptions related to the proportions of patients in the late and early stages of liver disease for HCC. Another limitation of our study was the general lack of data, especially in monitoring and HCC surveillance among adults with inactive or immune tolerant CHB. We tried to address this limitation by running sensitivity analyses.

The current study found regular monitoring of the thousands of adults diagnosed with inactive or immune tolerant CHB in the United States who are initially not recommended for treatment and subsequently receive antiviral therapy when indicated is cost-saving. The addition of HCC surveillance with the standard recommendation for biannual ultrasound and AFP would be cost-effective if the HCC risk is ≥ 0.55% per year such as patients with cirrhosis who have an estimated HCC incidence of over 3%/yr. A possible alternate lower cost HCC surveillance strategy consisting of only an annual ultrasound and biannual AFP would be cost-effective and could be considered for patients who have a lower HCC risk (≥ 0.3%/yr) including Asian men over 40 years and Asian women over 50 years.

## Supporting information

**S1 File.**
(DOCX)

**S1 Fig. Decomposition of total costs.** * "All other costs" are age-specific background medical costs (non-HBV-related, such as for heart disease).
(JPG)

**S2 Fig. Cost-effectiveness acceptability curve.** This figure shows the probability that each strategy might be cost-effective (Y-axis) at particular values of willingness-to-pay for QALYs (X-axis). Because the CP plus M&T strategy is highly likely to lead to both savings in costs and improvement in QALYs, it is highly likely that CP plus M&T is cost-effective, regardless of the willingness to pay for QALYs.
(JPG)

**S3 Fig. Tornado diagram.**
(JPG)

## Acknowledgments

**Disclaimer**: The findings and conclusions in this report are those of the authors and do not necessarily reflect the official position of the Centers for Disease Control and Prevention, or the authors' affiliated institutions.

## Author Contributions

**Conceptualization:** Mehlika Toy, David Hutton, Joshua A. Salomon, Samuel So.

**Data curation:** Mehlika Toy, David Hutton, Hang Pham, Joshua A. Salomon.

**Formal analysis:** Mehlika Toy, David Hutton, Samuel So.

**Funding acquisition:** Mehlika Toy, David Hutton, Erin E. Conners, Joshua A. Salomon, Samuel So.

**Investigation:** Mehlika Toy, David Hutton, Erin E. Conners, Hang Pham, Joshua A. Salomon, Samuel So.

**Methodology:** Mehlika Toy, David Hutton, Erin E. Conners, Hang Pham, Joshua A. Salomon, Samuel So.

**Project administration:** Mehlika Toy, David Hutton, Erin E. Conners, Hang Pham, Joshua A. Salomon.

**Resources:** Mehlika Toy, David Hutton, Erin E. Conners.

**Software:** Mehlika Toy, David Hutton.

**Supervision:** Mehlika Toy, David Hutton, Erin E. Conners, Joshua A. Salomon, Samuel So.

**Validation:** Mehlika Toy, David Hutton, Erin E. Conners, Hang Pham.

**Visualization:** Mehlika Toy, David Hutton.

**Writing – original draft:** Mehlika Toy.

**Writing – review & editing:** Mehlika Toy, David Hutton, Erin E. Conners, Hang Pham, Joshua A. Salomon, Samuel So.

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
