## [Decision Letter · Decision Letter 0]

8 Jan 2025

Cost-Effectiveness of Monitoring and Liver Cancer Surveillance among Patients with Inactive Chronic Hepatitis B

PONE-D-24-36839

Dear Dr. Toy,

We’re pleased to inform you that your manuscript has been judged scientifically suitable for publication and will be formally accepted for publication once it meets all outstanding technical requirements.

Kind regards,

Isabelle Chemin, PhD

Academic Editor

PLOS ONE

Additional Editor Comments (optional):

Reviewers' comments:

Reviewer's Responses to Questions

**Comments to the Author**

1. Is the manuscript technically sound, and do the data support the conclusions?

Reviewer #1: Yes

Reviewer #2: Yes

2. Has the statistical analysis been performed appropriately and rigorously? 

Reviewer #1: I Don't Know

Reviewer #2: Yes

3. Have the authors made all data underlying the findings in their manuscript fully available?

Reviewer #1: Yes

Reviewer #2: Yes

4. Is the manuscript presented in an intelligible fashion and written in standard English?

Reviewer #1: Yes

Reviewer #2: Yes

5. Review Comments to the Author

Reviewer #1: From a clinical perspective, this is a very important and immediate study. Due to the long-term nature and complexity of HBV infection, the inconsistency of guidelines for the diagnosis and treatment of hepatitis B, the accessibility of therapeutic drugs and the inconsistency of screening strategies in various regions of the world, the current treatment and control of CHB infection is extremely unsatisfactory. The significance of this study was used Markov model to figue out calculate the costs, health impact and cost-effectiveness, which can provide some reference evidence for the current inconsistent guidelines for treatment and screening strategies.

Reviewer #2: The paper is well-written and well-structured, with a thorough and well-executed statistical analysis in all aspects. Additionally, the references to existing literature appear complete. Therefore, I recommend accepting the paper as it is.

6. PLOS authors have the option to publish the peer review history of their article (what does this mean?). If published, this will include your full peer review and any attached files.

Reviewer #1: No

Reviewer #2: No

---

## [Editor Report · Acceptance letter]

10 Jan 2025

PONE-D-24-36839 

PLOS ONE

Dear Dr. Toy, 

I'm pleased to inform you that your manuscript has been deemed suitable for publication in PLOS ONE. Congratulations! Your manuscript is now being handed over to our production team.

Kind regards, 

on behalf of

Mrs Isabelle Chemin 

Academic Editor

PLOS ONE